# Unraveling the genetic architecture of congenital vertebral malformation with reference to the developing spine

Sen Zhao [1,2,3], Hengqiang Zhao[1,4,5], Lina Zhao[1,4,6], Xi Cheng[1], Zhifa Zheng[1,4,5], Mengfan Wu[7], Wen Wen[1], Shengru Wang[1], Zixiang Zhou [1], Haibo Xie[7], Dengfeng Ruan [8], Qing Li [1,4,5], Xinquan Liu[1], Chengzhu Ou [1], Guozhuang Li[1,4,5], Zhengye Zhao[1,4,5], Guilin Chen[1,4,5], Yuchen Niu[2,4,5,6], Xiangjie Yin[1,4,5], Yuhong Hu [1], Xiaochen Zhang[1], Deciphering disorders Involving Scoliosis and COmorbidities (DISCO) study*, Pengfei Liu[3,9], Guixing Qiu[1,2,4,5], Wanlu Liu [8], Chengtian Zhao [7], Zhihong Wu[2,4,5,6] ✉, Jianguo Zhang[1,2,4,5] ✉ & Nan Wu [1,2,4,5] ✉

Congenital vertebral malformation, affecting 0.13–0.50 per 1000 live births, has an immense locus heterogeneity and complex genetic architecture. In this study, we analyze exome/genome sequencing data from 873 probands with congenital vertebral malformation and 3794 control individuals. Clinical interpretation identifies Mendelian etiologies in 12.0% of the probands and reveals a muscle-related disease mechanism. Gene-based burden test of ultra-rare variants identifies risk genes with large effect sizes (*ITPR2*, *TBX6*, *TPO*, *H6PD*, and *SEC24B*). To further investigate the biological relevance of the genetic association signals, we perform single-nucleus RNAseq on human embryonic spines. The burden test signals are enriched in the notochord at early developmental stages and myoblast/myocytes at late stages, highlighting their critical roles in the developing spine. Our work provides insights into the developmental biology of the human spine and the pathogenesis of spine malformation.

Congenital vertebral malformation (CVM) refers to an abnormal morphology or segmentation of the vertebral column and can lead to progressive spinal deformity and cardio-pulmonary dysfunction. The incidence of CVM is around 0.13–0.50 per 1000 live births, which might be underestimated due to many asymptomatic individuals[1,2].

CVM may occur as an isolated phenotype, in association with multi-systemic anomalies (e.g., VACTR association, or Vertebral defects, Anal atresia, Cardiac defects, Tracheoesophageal fistula, and Renal anomalies), or within the phenotypic spectrum of chromosomal/Mendelian disorders (e.g., DiGeorge syndrome and Alagille

[1]Department of Orthopedic Surgery, Peking Union Medical College Hospital, Peking Union Medical College and Chinese Academy of Medical Sciences, Beijing 100730, China. [2]State Key Laboratory of Complex Severe and Rare Diseases, Peking Union Medical College Hospital, Peking Union Medical College and Chinese Academy of Medical Sciences, Beijing 100730, China. [3]Department of Molecular and Human Genetics, Baylor College of Medicine, Houston, TX 77030, USA. [4]Beijing Key Laboratory for Genetic Research of Skeletal Deformity, Beijing 100730, China. [5]Key laboratory of big data for spinal deformities, Chinese Academy of Medical Sciences, Beijing 100730, China. [6]Medical Research Center, Peking Union Medical College Hospital, Peking Union Medical College and Chinese Academy of Medical Sciences, Beijing, China. [7]Institute of Evolution & Marine Biodiversity, College of Marine Life Science, Ocean University of China, Qingdao 266003, China. [8]Zhejiang University-University of Edinburgh Institute (ZJU-UoE Institute), Zhejiang University School of Medicine, International Campus, Zhejiang University, 718 East Haizhou Road, Haining 314400, China. [9]Baylor Genetics, Houston, TX 77021, USA. *A list of authors and their affiliations appears at the end of the paper. ✉e-mail: orthoscience@126.com; jgzhang_pumch@yahoo.com; dr.wunan@pumch.cn

syndrome)[3,4]. In addition to classical Mendelian inheritance, we reported the compound inheritance of a *TBX6* null allele and a risk haplotype (T-C-A, rs2289292-rs3809624-rs3809627) in patients with hemivertebrae[5–7]. Interestingly, the risk T-C-A haplotype has a population frequency of around 25%[5], implicating the complex genetic architecture of CVM, where liable genetic variants may range from pathogenic mutations to common hypomorphic alleles.

The biological mechanism of CVM is also extremely heterogeneous, with more than 400 causal genes reported (hpo.jax.org/app/browse/term/HP:0003468). CVM-associated genes participate in various developmental and homeostatic processes of the skeletal system. For example, *TBX6* mediates the segmentation clock during somitogenesis, which gives rise to vertebrae and skeletal muscle[8]; *SOX9* and *RUNX2* are key transcription factors regulating the differentiation of cartilaginous and fibrous connective tissues[9]; *NOTCH2* is essential for the fundamental cellular processes and leads to gross developmental defects when mutated[10]. Identifying the specific molecular and pathological processes underlying each affected individual will promote the precise classification and clinical management of CVM. Conversely, mapping all CVM-associated genes back to the molecular network underlying the developing spine could shed light on the complex processes of embryogenesis.

To systematically investigate the genetic architecture and biological basis of CVM, we aggregated exome sequencing (ES) or genome sequencing (GS) data from 873 patients with CVM and 3794 control individuals. We firstly performed clinical interpretation to investigate Mendelian predispositions underlying CVM and further studied muscle-related CVM using the *alpk3a/b* double knockout (DKO) zebrafish model and *Alpk3*$^{-/-}$ mouse model. For undiagnosed cases, we performed gene-based burden tests on ultra-rare variants (URVs). We further performed single-nucleus RNA-seq (snRNA-seq) on the developing spine of human fetuses and integrated the expression data with genetic signals to gain insight into the biological relevance of CVM-associated genes. Here, we show the important role of muscle-related mechanisms in the pathogenesis of CVM. We report genetic association signals of ultra-rare variants (URVs) for CVM. We also demonstrate that the notochord at early stages and myoblast/myocytes at late stages of embryo development are enriched for the expression of CVM-associated genes.

## Results

### Cohort constitution

We included 873 probands diagnosed with CVM in our cohort (Table 1), which predominantly consists of patients exhibiting severe scoliosis or kyphosis, conditions necessitating surgical intervention. CVM occurred more frequently at the thoracic vertebrae than at the cervical and lumbar vertebrae (Supplementary Fig. 1). Intraspinal anomalies were frequent and predominantly affected females (Supplementary Table 1). Anomalies of the cardiac system, urogenital system, and digestive system represent the most prevalent extra-spinal defects (Supplementary Fig. 2), in accordance with the VACTR association[4,11]. After recruitment, ES or GS was performed on blood DNA from the probands and available parental samples (Supplementary Table 2).

### Mendelian predispositions and a muscle-related mechanism of CVM

We firstly performed clinical interpretation of ES/GS data and identified diagnostic variants in genes associated with Mendelian or chromosomal disorders in 105 (12.0%) probands (Fig. 1a). As anticipated, *TBX6*-associated congenital scoliosis (TACS) accounted for around one-half of all molecular diagnoses (Fig. 1a)[5,12]. The majority of TACS was caused by 16p11.2 deletion (Supplementary Data 1) and the others by *TBX6* variants (Supplementary Data 2). Aside from TACS, the rest of Mendelian etiologies underlying CVM presented extreme heterogeneity, with 36 individuals solved by 26 different disease genes (Fig. 1a and Supplementary Data 3). Dual molecular diagnoses were identified in two probands who presented mixed phenotypes of two Mendelian disorders (Supplementary Data 3). In addition, four pathogenic genomic deletions were identified in four patients with syndromic manifestations (Fig. 1a and Supplementary Data 4).

Most of the identified Mendelian disorders are associated with various forms of skeletal dysplasia or gross developmental disorders (Fig. 1a). Unexpectedly, we identified eight patients with pathogenic variants in genes associated with muscular disorders, including *MYH3*, *MYH7*, *ALPK3*, and *RYR1* (Fig. 1a and Supplementary Data 3). Muscular phenotypes of the eight patients were consistent with the Mendelian disease associated with each gene (Supplementary Data 5). In respect of CVM phenotypes, all eight patients presented segmentation defects (a.k.a., vertebral fusions) without structural anomaly of the vertebrae (Supplementary Data 5), whose prevalence is only 14.1% among all patients with CVM (Table 1). The skeletal fusion phenotype has been reported in *MYH3*-related disorders but not well-studied for the other muscle-related genes[13,14]. The expression of the four genes is predominantly enriched in myoblasts and myocytes in the embryonic spine (Supplementary Fig. 3).

To further investigate the role of muscle-related genes in vertebral malformations, we generated the *alpk3a/b* double knockout (DKO) zebrafish model and *Alpk3*$^{-/-}$ mouse model. The *alpk3a/b* DKO zebrafish model presented various vertebral segmentation malformations, including vertebral bar, truncated vertebrae, and blocked vertebrae (Fig. 1b). The low penetrance of vertebral malformation might be associated with gene compensation in zebrafish (Fig. 1c)[15]. For *Alpk3*$^{-/-}$ mouse mutants, we performed skeletal preparation or micro-CT at different weeks of age. Notably, we identified fusions of cervical vertebrae in adult but not postnatal mutants, and the proportion of vertebral fusion increased with age, suggesting a progressive malformation (Fig. 1d/e/f and Supplementary Table 3). Pathological analyses on *Alpk3*$^{-/-}$ mutants showed abnormal chondrogenesis and osteogenesis at the site of skeletal fusion, which also progressed with age (Supplementary Fig. 4). Consistently, long-term X-ray follow-up of the two patients with *ALPK3* variants also showed progressive fusions of the cervical vertebrae (Fig. 1d and Supplementary Fig. 5). Given the predominant expression of *ALPK3* in the muscle tissue (Supplementary Fig. 3), we suggest that the progressive vertebral fusion caused by

## Table 1 | Demographic and clinical profile of the study cohort

| Characteristics | CVM cohort |
|---|---|
| Number of probands, *n* | 873 |
| Sex, *n* of female (%) | 448 (51.3) |
| Age at diagnosis, yrs. (±SD) | 5.1 (6.0) |
| Vertebral malformation, *n* (%) | |
| Failure of formation | 301 (34.5) |
| Segmentation defect | 123 (14.1) |
| Mixed | 449 (51.4) |
| Intraspinal anomaly, *n* (%) | |
| Diastomyelia | 135 (15.5) |
| Syringomyelia | 81 (9.3) |
| Tethered spinal cord | 47 (5.4) |
| Multi-systemic anomaly, *n* (%) | |
| Cardiovascular | 95 (10.9) |
| Digestive | 62 (7.1) |
| Genitourinary | 65 (7.4) |
| Limbs | 42 (4.8) |
| Eyes and ears | 27 (3.1) |

*CVM* congenital vertebral malformation, *SD* standard deviation.

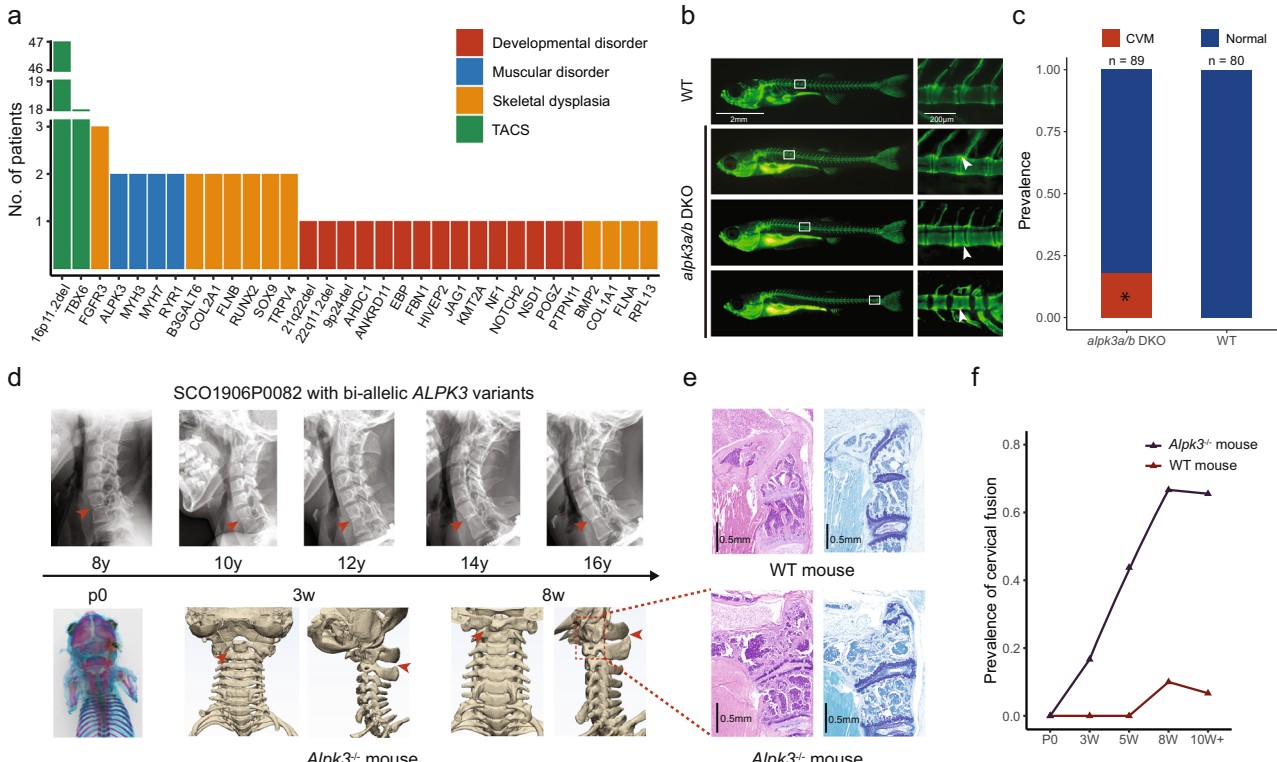

**Fig. 1 | Mendelian etiology in congenital vertebral malformation. a** Number of patients solved by variant(s) in each gene or genomic region. Disease genes were classified into four etiological categories. **b** Calcein staining on zebrafish at 23/24 dpf revealed various vertebral malformations, including vertebral bar, truncated vertebrae (middle), and blocked vertebrae (bottom). **c** Prevalence of congenital vertebral malformation (CVM) in wild-type (WT) or *alpk3a/b* double knock-out (DKO) zebrafish. *p* = 0.00016, two-sided Fisher's Exact test. **d** The progressive fusion of cervical vertebrae in SCO1906P0082 with bi-allelic *ALPK3* variants (lateral cervical spine X-ray) and *Alpk3*⁻/⁻ mouse (whole-mount skeletal staining or micro-CT). Arrowheads indicate the sites of vertebral fusion. **e** Cervical vertebrae pathology of *Alpk3*⁻/⁻ or WT mouse. H&E staining (left) and Toluidine blue staining of sagittal sections of cervical vertebrae revealed bony fusion of C1/C2 in *Alpk3*⁻/⁻ mouse. **f** The prevalence of vertebral fusion in *Alpk3*⁻/⁻ or mice at different ages (P0, newborn; w, weeks). Numbers of replicates are provided in Supplementary Table 3.

*ALPK3* might be associated with local mechanical changes of the paraspinal muscle which induce osteogenesis.

## Gene-based burden analysis of ultra-rare variants

Upon identifying monogenic etiology in 12.0% of patients, we next sought for genetic factors that contribute to CVM with reduced penetrance or via multifactorial mechanisms. We performed a gene-based mutational burden analysis of URVs in the 744 molecularly undiagnosed patients against 3740 in-house controls that passed quality control (QC) (Supplementary Table 4).

Five genes reached a false discovery rate (FDR) < 0.05 after multiple-test adjustment (Fig. 2a, Supplementary Data 6). The top signal *ITPR2* harbored one loss-of-function[14] variant and twenty-five missense/inframe variants in twenty-six patients with CVM (odds ratio [OR] = 3.83, FDR = 0.0088, Supplementary Data 6 and 7). *ITPR2* encodes an inositol 1,4,5-triphosphate receptor that mediates intracellular calcium release and regulates osteoclast differentiation via activation of the IRE1α/XBP1 pathway[16]. Single nucleotide polymorphisms (SNPs) in *ITPR2* have been associated with Kashin−Beck disease, a chronic osteochondropathy characterized by cartilage degeneration[17]. The expression of *ITPR2/ltpr2* is enriched in chondrocyte progenitors/cartilage primordium of the embryonic spine of humans (Supplementary Fig. 6) and mice (Fig. 2b), suggesting its potential role in the chondrogenesis of the vertebral column. Variants identified in CVM patients are enriched in the IP3 binding domain of ITPR2 (Fig. 2c, d), whose homolog in *ITPR1* is enriched for pathogenic variants associated with congenital ataxia[18]. Therefore, the impairment of IP3 binding affinity might underlie *ITPR2*-related CVM.

*SEC24B* is another notable signal with nine URVs identified in CVM patients (OR = 5.77, FDR = 0.034, Supplementary Data 6 and 7), and is widely expressed in the embryonic spine (Supplementary Fig. 6). SEC24B mediates the endoplasmic reticulum-to-Golgi transportation of VANGL2, a core planar cell polarity (PCP) gene essential for the development of the neuro tube and the spine[19]. The *Sec24b*⁻/⁻ mouse is embryonic lethal and presents in-uterus tail malformation (mouse-phenotype.org/data/genes/MGI:2139764)[20], which is reminiscent of the vangl2 *loop tail* mutant and mimics the CVM phenotype in humans[21]. Through an in-house gene-matching, we identified another missense variant in *SEC24B* which arose de novo in a patient with idiopathic short stature. Upon follow-up, spina bifida was identified in this patient, implicating the hypoplasia of the spine and the neural tube that are potentially associated with the *SEC24B* variant (Supplementary Fig. 7). Among the other three exome-wide significant genes, *H6PD* encodes the hexose-6-phosphate dehydrogenase which is important for maintaining muscle homeostasis[22] and might be associated with CVM through the muscular mechanism similar to that of *ALPK3*. *TPO* encodes a thyroid peroxidase, whose etiological relation with CVM is not clear. *TBX6* showed a significant association despite the exclusion of patients diagnosed with TACS, which is probably driven by variants of uncertain significance that did not fulfill a molecular diagnosis (Supplementary Data 6 and 7).

We then performed a pathway enrichment analysis on all 353 nominally significant (*p* < 0.05) genes from the burden test (i.e., CVM-associated genes). The most enriched Gene Ontology (GO, geneontology.org) processes included morphogenesis of various organs and apoptosis during development (Fig. 2e), consistent with CVM being a developmental defect. Among significantly enriched molecular

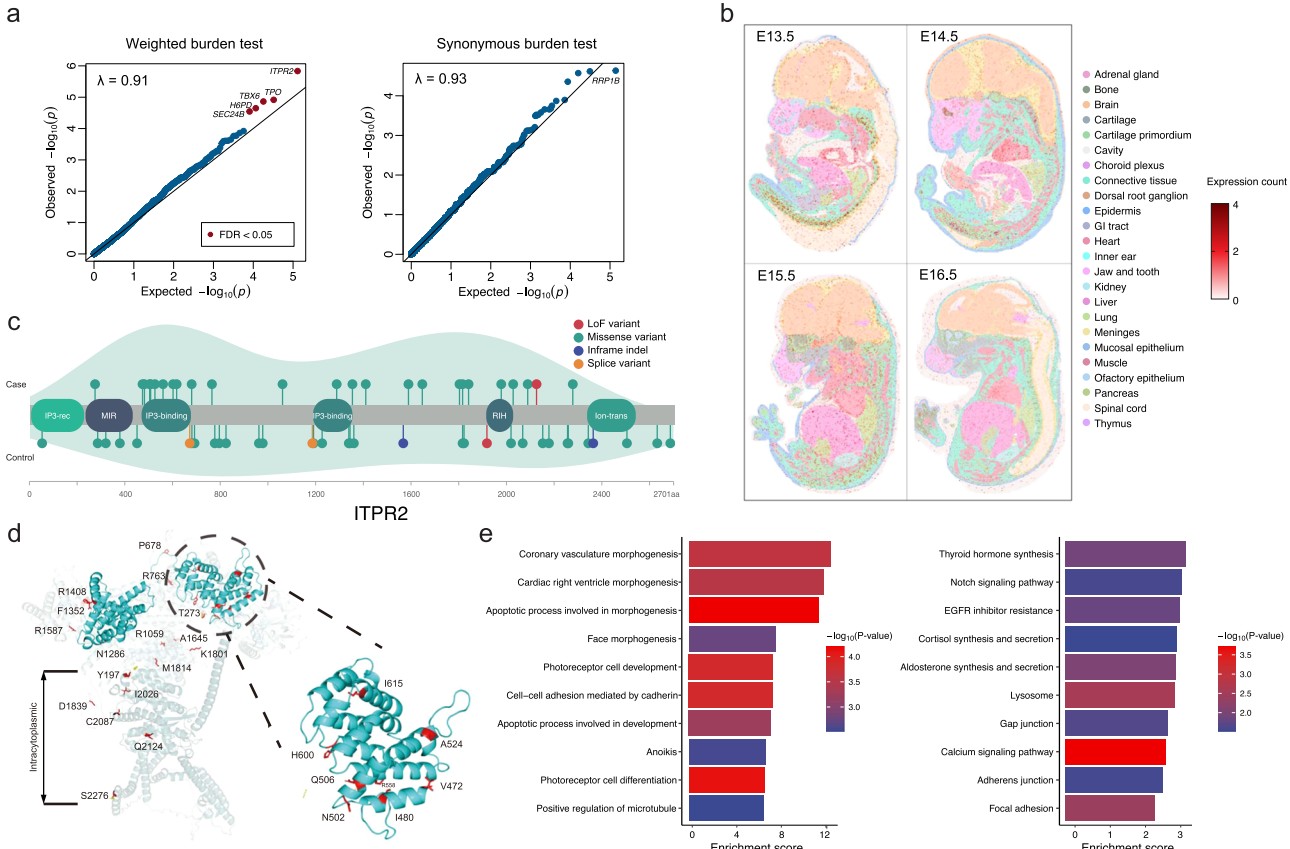

**Fig. 2 | Gene-based burden analysis of ultra-rare variants. a** Quantile-quantile plots of gene-based burden analysis. Burden test was performed using a two-sided aggregated Cauchy association test (ACAT). The Storey-Tibshirani procedure was used to adjust for multiple tests. Genes with exome-wide significance (FDR < 0.05) were labeled in red (left). The burden test of synonymous variants suggests that the pipeline was well-calibrated. Expected *p*-values were calculated by randomly switching case-control labels (permutation *n* = 1000). **b** Spatial-temporal expression of *Itpr2* in mouse embryos from E13.5 to E16. **c** Distribution of ultra-rare variants on ITPR2 protein. The density curves were created using the ggplot2 package in R. Domain information was obtained from Uniprot (http://www.uniprot.org/). **d** Mapping of *ITPR2* variants on its 3D structure, with the IP3-binding domain featured. **e** Pathway enrichment analysis of nominally significant burden signals. One-sided hypergeometric tests were used without adjustment for multiple tests. GO Gene Ontology, KEGG Kyoto Encyclopedia of Genes and Genomes.

pathways documented in the KEGG database (genome.jp), we highlight the 'calcium signaling pathway' which had a *p*-value of 0.0002 (Fig. 2e). Calcium signaling is essential for the development and homeostasis of the musculoskeletal system[23]. CVM-associated genes involved in this pathway included *ITPR2*, *PHKA1*, *MYLK3*, *PDGFRA*, *ADCY9*, *ATP2B1*, and *CACNA1S* (KEGG database: hsa04020)[24]. In addition, among the Mendelian genes for CVM, *RYR1* is also involved in the calcium signaling pathway.

## The role of CVM-associated genes in the developing spine

To further explore the role of CVM-associated genes in the developing spine, we performed snRNA-seq on the human embryonic spine at 5w, 6w, 7w, 8w, and 17w of gestational age. We identified and annotated 30 clusters of skeletal, muscular, neural, and miscellaneous cells (Fig. 3a and Supplementary Data 8). The decreasing proportion of progenitor cells and increasing proportion of differentiated cells along the gestation stages is in concordance with the embryonic developmental process (Fig. 3b).

We examined the aggregated expression of nominally significant genes from the burden test (*n* = 353) among various cell types at different developmental stages using the EWCE package[25]. The expression of burden signals was enriched in the notochord cells, which only existed in early-stage samples (5w, 6w, and 7w, Fig. 4). The notochord represents an essential structural element of the body axis and a scaffold for spine development[6–8]. Disruption of the notochord caused by the depletion of *col8a1* or *dstyk* has been shown to cause CVM in

zebrafish models[26,27]. Burden signals that were highly expressed in the notochord include *PIK3C2G*, *NTN1*, and *VIM* (Supplementary Fig. 8). Myoblasts/myocytes showed modest enrichment in later stages of embryonic development (7w, 8w, and 17w), reiterating the myogenic mechanism of CVM (Fig. 4).

## Discussion

The immense locus heterogeneity, compounded by the complexity of the genetic architecture of CVM, challenges the delineation of the disease etiology. In this study, we provide insights into the complex genetic architecture through different lenses: medical genetic interpretation for Mendelian etiology and gene-based URV burden test for oligogenic effect.

The Mendelian etiology explained 12.0% of the patients with CVM, which is similar to the diagnostic yield reported by us and others[3,28]. Attributed to the large sample size and agnostic sequencing method (ES/GS), our molecular diagnoses provide a comprehensive and unbiased demonstration of locus heterogeneity. In addition to the diagnoses of various skeletal dysplasia and gross developmental defects, we unexpectedly identified eight cases with muscular disorders associated with four disease genes. The CVM phenotype associated with muscular disorders tends to be vertebral fusion rather than morphological anomalies of the vertebrae. *Alpk3*$^{-/-}$ mouse model and *alpk3a/b* DKO zebrafish model recapitulated the human phenotype. *Alpk3*$^{-/-}$ mouse model further revealed that muscle-related vertebral fusion progresses with age and is caused by

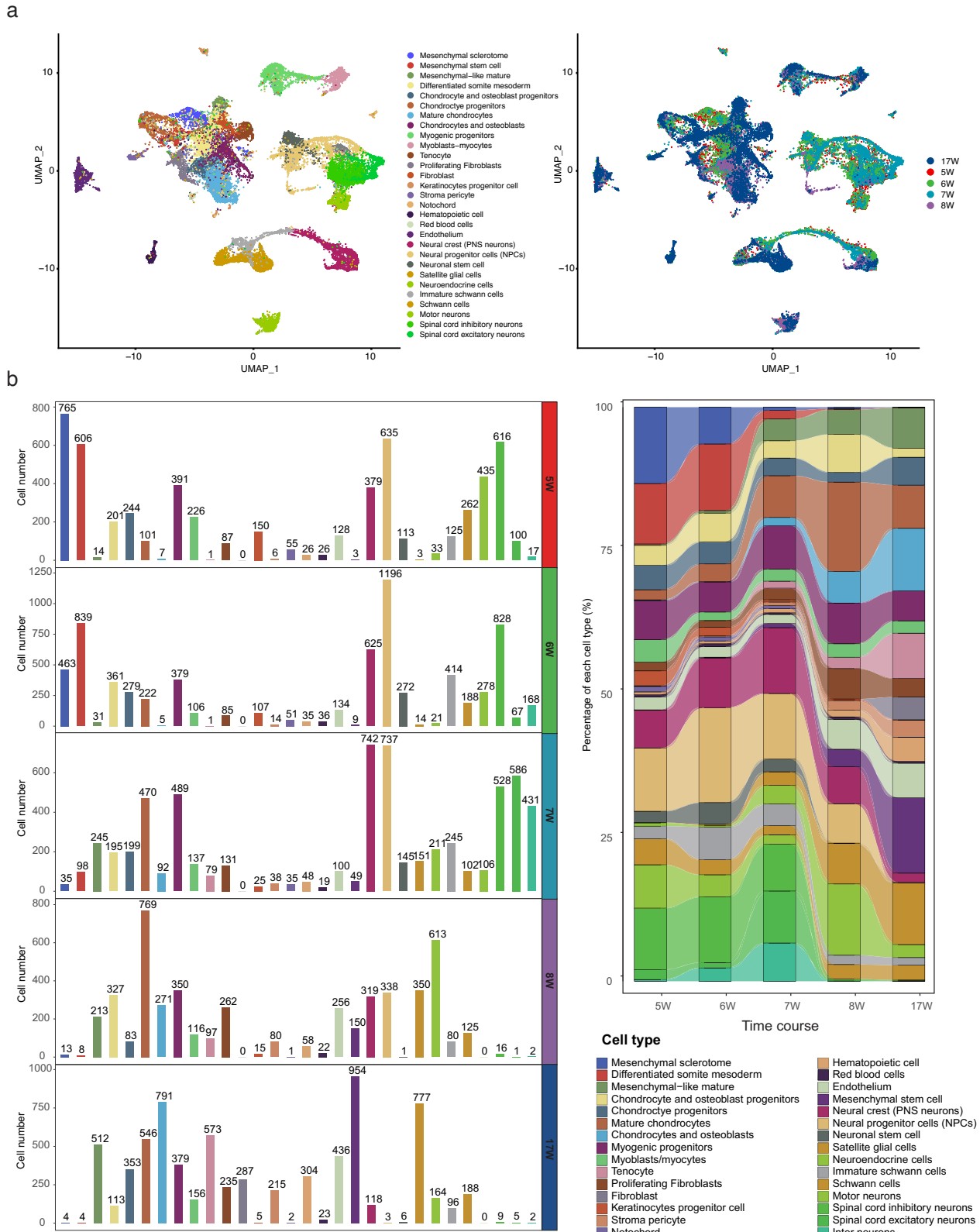

**Fig. 3 | Single-cell transcriptome of the human embryonic spine. a** Uniform manifold approximation and projection (UMAP) visualization of single cells grouped by annotated cell clusters or gestational age. **b** The cell number and the proportion of different cell types at each developmental stage.

abnormal osteogenesis at the C1/C2 junction. Based on current findings, we propose that muscular disorders cause vertebral fusion by inducing mechanical changes in small paraspinal muscles, which is in line with the hypothesis previously raised in a mouse study of the *MYH3*-related skeletal fusion[29]. Distinguishing muscle-related

vertebral fusion from bona fide malformation of the vertebral column is important for genetic counseling, diagnosis, and disease management. The progressive nature of this distinct clinical entity also prompts the opportunity for early interventions that target myoblasts or myocytes.

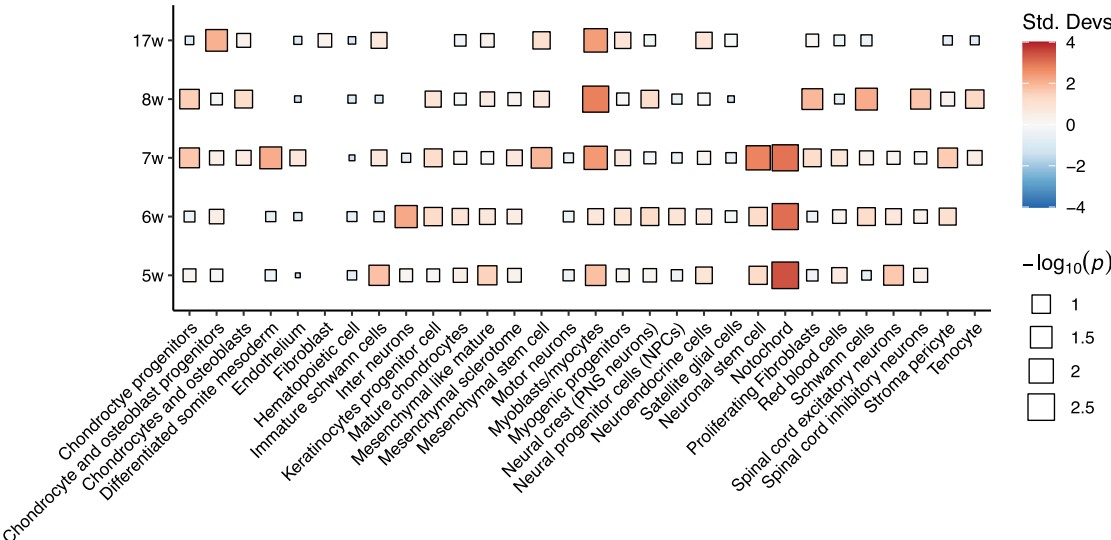

**Fig. 4 | Enrichment of genetic association signals in the developing spine.** The enrichment of mutational burden signals in each cell type at each developmental stage. The enrichment analyses were performed using the EWCE package in R. *p*-values were calculated using one-sided bootstrap tests (*n* = 1000) without adjustment for multiple tests. Std Devs standard deviations from the mean.

## Methods

### Ethical statement

The human genetic study was approved by the Ethics Committee of Peking Union Medical College Hospital (JS-908). Written informed consent was obtained from all participants. Participants were not provided with financial compensation in this study.

The collection of human embryos underwent a rigorous ethical review process and was approved by the Ethics Committee of Peking Union Medical College Hospital (I-22PJ819). We provided all donors with comprehensive information about the nature, purpose, and potential outcomes of the research, ensuring that they were fully aware of and understood the aims of our study. We ensured that every donor signed a voluntary consent before any embryo collection occurred during their legally approved pregnancy termination. Donors were not provided with financial compensation in this study. All procedures were carried out strictly following the guidelines of 'Management of Human Genetic Resources', as stipulated by the Ministry of Science and Technology of China (no. 717, effective from July 1, 2019). Detailed records of the embryo acquisition, consent verification, and ethical review were securely maintained by the research team.

Zebrafish experiments were carried out in compliance with conventional animal handling protocols and received approval from the Animal Care Committee at the Ocean University of China (OU-2012316). Mouse studies were approved by the Ethics Committee of Peking Union Medical College Hospital (XHDW-2022-030).

### Cohort recruitment

We consecutively enrolled Chinese individuals (families) affected with CVM who underwent spinal surgery at Peking Union Medical College Hospital from November 2012 to November 2021 for correction of scoliosis or kyphosis under the framework of Deciphering disorders Involving Scoliosis and Comorbidities (DISCO) study. All individuals underwent a physical examination, spinal X-ray, spinal computed tomography, spinal magnetic resonance imaging, echocardiography, and renal ultrasound. The diagnosis of CVM was confirmed by both a radiologist and an orthopedic surgeon. The control cohort was aggregated from individuals who underwent exome sequencing at Peking Union Medical College Hospital, for clinical or research purposes. Individuals recorded to have skeletal malformation or other

A gene-based collapsing strategy allowed us to identify the disease liability of URVs with adequate statistical power. The top signal *ITRP2*, together with other nominally significant signals, pinpointed the calcium signaling pathway underlying the pathogenesis of CVM. Calcium signaling is essential for the development of both skeletal and muscular systems. In the chondrocytes, cytosolic calcium oscillation well-tunes downstream signaling molecules such as SOX9[30]. In skeletal muscle development, the spatiotemporal pattern of the Ca2+ signal is vital for myofibril organization and sarcomere assembly[31]. Pathway enrichment analysis of URV association signals showed various morphogenesis and apoptosis processes, consistent with CVM being a developmental disease.

The development of the spine is a multi-stage and highly regulated process. We performed snRNA sequencing on human embryos to further understand the role of disease-associated genes in the developing spine. Expression enrichment analysis of genetic association signals enabled us to locate the key stages and cell types of the developing spine without manipulating the embryo, which is inapplicable in humans. Here we used human as the 'model organism' and performed integrated analyses between genetics association signals and single-cell transcriptome of the embryonic spine. We found that URV genetic associations concordantly ascribed the disease liability to the notochord during the early developmental stages. The notochord plays a central role in vertebral patterning[32]. In zebrafish models, notochord defects of the embryo are spatially correlated with vertebral malformations in the adults, implicating a causal relationship between notochord disruption and CVM[26,27]. Our findings uncovered the potential causal role of notochord in human CVM through a 'forward genetics' strategy. The enrichment in myocyte/myoblasts was identified at the later developmental stages, which is consistent with the late-onset and progressive vertebral fusion associated with Mendelian muscular disorders identified in our cohort.

Overall, we delineated the genetic basis of CVM from Mendelian predisposition to rare variant architecture through cohort-based exome/genome sequencing analyses. By incorporating genetic signals with snRNA-seq of human embryos, we revealed the key developmental stages and cell types of the developing spine.

congenital anomalies were excluded. The retained control cohort included 3794 unrelated individuals. The sex of the participants was determined based on self-report and validated through genetic data. No sex-specific analysis was performed because CVM is not a sex-biased disease.

## Exome sequencing (ES) and genome sequencing (GS)

Exome sequencing (ES) or genome sequencing (GS) was performed on peripheral blood DNA from the participants and available parental samples. The number of probands (families) and control samples who underwent ES or GS are shown in Supplementary Table 2. Exome sequencing was performed using nine different capture kits with an average depth of around 70X. PCR-free genome sequencing was performed under an average depth of around 35X (Supplementary Methods).

Raw sequencing data were processed using the Peking Union Medical College Hospital pipeline (PUMP)[3], generating a genomic variant call file (gVCF) for each individual (Supplementary Methods). Individual gVCFs were combined and jointly called using the Sentieon® software (v202010), followed by variant-level and sample-level quality controls (Supplementary Methods). Single nucleotide variants (SNVs) and insertion/deletions (indels) were annotated using variant effect predictor (https://useast.ensembl.org/info/docs/tools/vep/index.html, v105).

## Identification of causal pathogenic variants

We performed clinical analysis of ES or GS data to identify Mendelian disorders underlying CVM. For SNV/indels, we screened rare variants (gnomAD allele frequency ≤ 1%) for potentially diagnostic variant(s), i.e., pathogenic/likely pathogenic variant(s) that could explain the clinical landscape of an individual (Supplementary Methods). For individuals diagnosed with *TBX6*-associated congenital scoliosis (TACS), i.e., those who carry *TBX6* null alleles or 16p11.2 deletions, the haplotyping of the T-C-A (rs2289292, rs3809624, rs3809627) haplotype was performed as in our previous study[3].

Copy number variants (CNVs) were called from ES and GS bam files separately. Known pathogenic CNVs that could explain patient phenotypes were classified as diagnostic variants. Detailed processes of CNV calling and interpretation are provided in Supplementary Methods.

## Gene-based burden analysis of ultra-rare variants

Gene-based burden analyses were performed on ultra-rare variants (URVs), defined by a gnomAD control population-max allele frequency (MAF) ≤ 0.01% and a cohort allele count ≤3. The retained variants were annotated with transcript-level information according to GENCODE v39 (https://www.ensembl.org/). If a variant is allocated to multiple GENCODE transcripts of a single gene, the transcript labeled as 'canonical' by GENCODE was selected. We then assigned weight values (ranging from 0 to 1) to each variant based on the variant type and bioinformatic prediction results (Supplementary Table 5). The mutational burden of a given gene in an individual was assigned as the maximum weight value among all ultra-rare variants in the gene. Only genes with at least ten qualified variants (weight ≠ 0) across the cohort were processed for the burden test. The weighted mutational burden test on each gene was performed using the ACAT package (v0.91)[33]. Expected *p*-values were calculated by randomly switching case-control labels (permutation *n* = 1000). The Storey-Tibshirani procedure was used to adjust for multiple tests and generate a false-positive rate (FDR).

## Pathway enrichment analysis

Genes with nominal significance from burden test (*n* = 353) were processed for pathway enrichment analysis. The 'enrichGO' and enrichKEGG functions from clusterProfiler package (v4.0)[24] were used to test the enrichment of biological pathways documented in the Gene Ontology (GO, geneontology.org) and Kyoto Encyclopedia of Genes and Genomes (KEGG, www.genome.jp) databases. *P*-values were calculated using the hypergeometric test. The enrichment score was defined as the ratio of positive hits in each pathway controlled against the size of the pathway.

## alpk3a and alpk3b double knock-out (DKO) zebrafish model

Frameshift variants in exon2 of *alpk3a* and exon3 of *alpk3b* were generated in the AB/TU zebrafish strain using the CRISPR-Cas9 system (Supplementary Methods). Knock-out of *alpk3a* and *alpk3b* was confirmed via quantitative real-time PCR (qRT-PCR). *alpk3a*$^{-/-}$ and *alpk3b*$^{-/-}$ zebrafish strains were crossed and bred to generate the *alpk3a/b* DKO strain.

For calcein staining, zebrafish at 23/24 dpf were kept alive and soaked in 0.2% Calcein (Sigma) solution with a pH of 7.5 for 15 min, followed by a double rinse with system water. The zebrafish, stained in vivo, were then anesthetized using a 0.01% concentration of tricaine methanesulfonate (MS222), and positioned in a 3% methyl cellulose medium for imaging. The imaging process was carried out using a Leica M165FC fluorescence microscope. The occurrence of vertebral malformation between KO and WT groups was tested using a two-sided Fisher's Exact test. Number of replicates: DKO = 89, WT = 80. Investigators were blinded to group allocation during data collection and analysis. Sex was not considered in our study design because CVM is not a sex-biased disease.

## Alpk3$^{-/-}$ mouse model

Exon3 deletion of *Alpk3* was induced in the C57BL/6 mouse strain using the CRISPR-Cas9 system (Supplementary Methods). Knock-out of *Alpk3* was confirmed via Data-independent acquisition mass spectrometry (DIA-MS, Orbitrap Exploris™ 240 Mass Spectrometer) at the protein level.

Whole-mount skeletal preparation was performed on wild-type and *Alpk3*$^{-/-}$ mouse at P0 (newborn) according to the standard protocol[34]. Number of replicates: WT = 8, KO = 14. Micro-CT was performed on wild-type and *Alpk3*$^{-/-}$ mice at 3w, 5w, 8w, and 10w of age using the PerkinElmer® Quantum FX machine. Numbers of replicates are specified in Supplementary Table 3. Mimics software (v19.0) was used for the 3D reconstruction of micro-CT data. Histological analyses were performed on wild-type and *Alpk3*$^{-/-}$ mice at 5w and 8w. Sagittal sections at C1/C2 junction were obtained and underwent decalcification. After decalcification, H&E and Toluidine blue staining were performed (Supplementary Methods)[35]. Five replicates each for WT and KO at 5w and 8w were performed for histological analyses. Investigators were blinded to group allocation during data collection and analysis. Sex was not considered in our study design because CVM is not a sex-biased disease.

## Single-nucleus RNA sequencing and spatial transcriptome of the growing spine

Human fetuses at 5w, 6w, 7w, 8w, and 17w gestational weeks from elective pregnancy termination were collected (*n* = 1 for each developmental stage). Details of tissue isolation, library preparation, snRNA-seq, and raw data processing were described in Supplementary Methods. The enrichment analyses of genetic signals in snRNA-seq data were performed using the EWCE software (v1.6)[25]. We tested nominally significant genes from the burden test. The enrichment of gene expression across cell types was tested with *n* = 1000 bootstraps against random background genes.

Spatial transcriptome data of mouse embryos were obtained from a recent study[36] (https://db.cngb.org/stomics/mosta/, accession date 2022/02/14). Annotated data (.h5ad) from E13.5, E14.5, E15.5, and E16.5 were used for visualization of spatial-temporal expression of candidate genes.

**Reporting summary**

Further information on research design is available in the Nature Portfolio Reporting Summary linked to this article.

## Data availability

The raw genetic sequencing data for CVM patients and control individuals generated in this study have been deposited in the Genome Sequence Archive (GSA, https://ngdc.cncb.ac.cn/gsa-human/)[37] under accession numbers HRA006007 and HRA006052. Raw snRNAseq data generated in this study have been deposited in the GSA under the accession number HRA006073. All raw sequencing data deposited in GSA are under restricted access and only academic use would be approved. A response would be expected within a week. The reference genome used in this study is genome assembly GRCh37/hg19 (https://www.ncbi.nlm.nih.gov/datasets/genome/GCF_000001405.13/). Public data repositories employed throughout this paper include GENCODE (hg19, https://www.gencodegenes.org/human/release_39.html), Ensembl (hg19, https://useast.ensembl.org/index.html), and Genome Aggregation Database (gnomAD v2.1.1, https://gnomad.broadinstitute.org/). Source data are provided with this paper.

## Code availability

The codes used for burden analysis are deposited on Github (github.com/zhq921/Burden-pipeline).

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

## Acknowledgements
We thank Ms. Yufei Dong for her help in producing the figures. This research was funded by the National Key Research and Development Program of China (2022YFC2703100 to N.W. and Z.W.), CAMS Innovation Fund for Medical Sciences (CIFMS, 2021-I2M-1-051 to T.Z. and N.W., 2021-I2M-1-052 to Z.W., 2020-I2M-C&T-B-030 to T.Z.), The National Natural Science Foundation of China (82072391 to N.W., 81930068 and 81772299 to Z.W., 81972037 and 82172382 to T.Z., 81972132 to G.Q.), National High-Level Hospital Clinical Research Funding (2022-PUMCH-D-004, 2022-PUMCH-C-033), The Non-profit Central Research Institute Fund of Chinese Academy of Medical Sciences (No. 2019PT320025 to N.W.).

## Author contributions
N.W., J.Z., Z.W., P.L., G.Q. and S.W. designed the study. Z.Zha., G.L., X.Y., G.C. and Y.N. collected and processed blood samples. S.Z. and H.Z. performed genetic association analyses. X.C., W.W., X.L., C.O., Y.H, X.Z. and Z.Zho. performed medical genetics analyses. Z.Zhe. performed mouse experiments. M.W., H.X. and C.Z. performed zebrafish experiments. L.Z. and Y.N. performed single-cell RNAseq library preparation. D.R., Q.L. and W.L. analyzed single-cell RNAseq data. S.Z. drafted the paper. All authors contributed to the final version of this paper. S.Z., H.Z., L.Z. and X.C. contributed equally as first authors. J.Z., Z.W. and N.W. contributed equally as senior authors.

## Competing interests
The authors declare no competing interests.

## Additional information

## Deciphering disorders Involving Scoliosis and COmorbidities (DISCO) study

Sen Zhao [1,2,3], Hengqiang Zhao[1,4,5], Lina Zhao[1,4,6], Xi Cheng[1], Zhifa Zheng[1,4,5], Wen Wen[1], Shengru Wang[1], Zixiang Zhou [1], Qing Li [1,4,5], Xinquan Liu[1], Chengzhu Ou [1], Guozhuang Li[1,4,5], Zhengye Zhao[1,4,5], Guilin Chen[1,4,5], Yuchen Niu[2,4,5,6], Xiangjie Yin[1,4,5], Yuhong Hu [1], Xiaochen Zhang[1], Guixing Qiu[1,2,4,5], Zhihong Wu[2,4,5,6 ✉], Jianguo Zhang[1,2,4,5 ✉], Nan Wu [1,2,4,5 ✉], Sen Liu[1,2], Zihui Yan[1,2], Xiaoxin Li[6], Bowen Liu[1,2], Yingzhao Huang[1,2], Guangxi Gao[6], Qing Liu[6], Jianle Yang[1,2], Xinyu Yang[1,2], Aoran Maheshati[1,2], Jihao Cai[1,2], Yuanpeng Zhu[1,2], Jie Wang[1,2], Yang Yang[1,2], Ziquan Li[1,2], Guanfeng Lin[1,2] & Xiaohan Ye[1,2]

