## [Peer Review File · Nature Communications]

Unraveling the genetic architecture of congenital vertebral malformation with reference to the developing spineREVIEWER COMMENTS

Reviewer #1 (Remarks to the Author):

The manuscript of "Unraveling the genetic architecture of congenital vertebral malformation with reference to the developing spine" from Zhao et al. conducted comprehensive genetic analyses in a large cohort (873 probands) of congenital vertebral malformation (CVM) and presented the genetic architecture for this developmental disorder. In addition to showing 12% Mendelian etiologies, the authors also revealed several novel CVM-associated genes. Their further experimental evidence from the exome-wide association study and embryonic single-nucleus transcriptome also supported a portion of polygenic architecture for CVM. This manuscript is informative for the clinical and basic researches of CVM.

Major comments:

(1) Figure 3d is misleading:

The portions of "Mendelian", "TACS", and "CNV" are accurately overlapping (instead of being independent). TACS is Mendelian, and the majorities of TACS are also CNV-associated.

2) TBX6 variants:

Only null variants are identified and shown in Supplementary Table 4. Is there any pathogenic missense variant of TBX6 (functionally acting as a LoF allele) in this cohort?

Minor comments:

(a) The genotype "-/-" symbols should be shown as superscripts.

(b) Are the genetic variants shown in Supplementary Table 5 are heterozygous or homozygous? Please show this information in the table. Also, please explain P and LP for the readers.

(c) It is suggested to use capitalized WT (instead of wt) for wild-type in the text and figure legends.

(d) The information of the following references is incomplete: Refs. 6, 10, and 29. Please provide issue/page/article numbers.

Reviewer #2 (Remarks to the Author):

The authors conducted a thorough investigation into the genetic and developmental etiologies of congenital vertebral malformation (CVM) using various approaches, including gene-based burden tests, exome-wide association studies, and single-nucleus RNAseq techniques on the human embryonic spine. As a result, they identified a set of genetic regions and genes, such as ITPR2, TBX6, TPO, H6PD, and SEC24B, as risk factors for CVM. While the study is well-designed and has significant clinical implications, its primary significance lies in advancing our comprehension of the genetic underpinnings of CVM.

Major comments:

As also explained by the authors, CVM can manifest through a variety of mechanisms, including gene aberrations responsible for skeletal development and anomalous mechanical stresses stemming from muscular deformities (e.g., MYH3 mutant-related phenotype). Consistently, the authors have identified genetic factors that may be linked to these mechanisms in humans. These results were largely anticipated (largely by studies using other vertebrate species), and unfortunately, I was unable to

discern the broader implications of these findings. For example, what constitutes the novel "insights into the developmental biology and pathology of the human spine"?

Minor comments:

Figure 2e

The exclusive reliance on p-values in GO term enrichment or KEGG analysis is a widespread statistical misconception. It is crucial to present effect sizes in conjunction with statistical significance (lower than alpha level you have settled for this study), as the number of GO terms or KEGG pathways covered by each term/pathway can exhibit significant variations. The p-value, on its own, primarily reflects statistical accuracy, while effect sizes offer a quantitative measure of the practical significance or magnitude of the observed effects for each term or pathway.

I.178 - 183

I assume the result could change if the above statistical misconception was solved.

Figure 3c

Same as Figure 2e.

I.195 - 198

The authors devoted considerable effort to gather the comprehensive dataset presented in this manuscript, and I commend their dedication and diligence. However, this is the main reason why I am concerned with the description and results of current version. Relying only on P-values works fine for EWAS or GWAS analysis, as the sample number used is the same for every locus, however, this does not stand for GO term and KEGG pathway analyses as I explained above.

Figure 4 and Methods

Please specify how many embryos were used for each developmental stage. While statistical power and accuracy may be compromised if all of these were done only for one sample each ($n=1$), it is still precious data.

Response letter

Reviewer #1 (Remarks to the Author):

The manuscript of “Unraveling the genetic architecture of congenital vertebral malformation with reference to the developing spine” from Zhao et al. conducted comprehensive genetic analyses in a large cohort (873 probands) of congenital vertebral malformation (CVM) and presented the genetic architecture for this developmental disorder. In addition to showing 12% Mendelian etiologies, the authors also revealed several novel CVM-associated genes. Their further experimental evidence from the exome-wide association study and embryonic single-nucleus transcriptome also supported a portion of polygenic architecture for CVM. This manuscript is informative for the clinical and basic researches of CVM.

Major comments:

(1) Figure 3d is misleading:

The portions of “Mendelian”, “TACS”, and “CNV” are accurately overlapping (instead of being independent). TACS is Mendelian, and the majorities of TACS are also CNV-associated.

Response: Thank you for pointing this out. Indeed, SNV diagnosis, CNV diagnosis and TACS can all be classified into the ‘Mendelian’ category. We have revised Figure 3d accordingly.

2) TBX6 variants:

Only null variants are identified and shown in Supplementary Table 4. Is there any pathogenic missense variant of TBX6 (functionally acting as a LoF allele) in this cohort?

Response: That is an important point. We did publish our functional screening of missense variants in TBX6¹. However, the functional assays we performed in that study were still preliminary and we do not yet have a clinical-level functional assay or deep mutation scanning (DMS) to ‘diagnose’ patients with potential pathogenic missense variants in TBX6. Therefore, patients with non-truncating variants in TBX6 were still treated as ‘undiagnosed’ in this study and went through association analysis.

Minor comments:

(a) The genotype “-/-” symbols should be shown as superscripts.

Response: Thank you for pointing this out. We have revised the symbols accordingly throughout the manuscript and figures.

(b) Are the genetic variants shown in Supplementary Table 5 are heterozygous or homozygous? Please show this information in the table. Also, please explain P and LP for the readers.

Response: All the variants in Supplementary Table 5 are heterozygous. We have added an additional column to show that and have explained P and LP in the table legend.

(c) It is suggested to use capitalized WT (instead of wt) for wild-type in the text and figure legends.

Response: Thank you for the suggestion. We have revised wt to WT throughout the manuscript and the figures.

(d) The information of the following references is incomplete: Refs. 6, 10, and 29. Please provide issue/page/article numbers.

Response: Thank you for pointing this out. We have updated the issue information for these references.

Reviewer #2 (Remarks to the Author):

The authors conducted a thorough investigation into the genetic and developmental etiologies of congenital vertebral malformation (CVM) using various approaches, including gene-based burden tests, exome-wide association studies, and single-nucleus RNAseq techniques on the human embryonic spine. As a result, they identified a set of genetic regions and genes, such as *ITPR2*, *TBX6*, *TPO*, *H6PD*, and *SEC24B*, as risk factors for CVM. While the study is well-designed and has significant clinical implications, its primary significance lies in advancing our comprehension of the genetic underpinnings of CVM.

Major comments:

As also explained by the authors, CVM can manifest through a variety of mechanisms, including gene aberrations responsible for skeletal development and anomalous mechanical stresses stemming from muscular deformities (e.g., *MYH3* mutant-related phenotype). Consistently, the authors have identified genetic factors that may be linked to these mechanisms in humans. These results were largely anticipated (largely by studies using other vertebrate species), and unfortunately, I was unable to discern the broader implications of these findings. For example, what constitutes the novel "insights into the developmental biology and pathology of the human spine"?

Response: Thank you for raising this important point. We agree that the biological mechanisms we proposed in this study are largely anticipated based on previously studied vertebrate models. We believe our major novelty compared with previous studies is that we used humans as the 'vertebrate model' in genetic and embryonic transcriptome analysis. Among various established pathogenesis mechanisms and hypotheses, we identified those that play important roles in human spinal malformations.

For example, it has been controversial whether *MYH3* mutations cause human skeletal malformation through a muscle-related or bone-related mechanism^{2,3}. With the identification of mutations in other muscle-related genes such as *ALPK3* and *RYR1* from our study, it has become clearer that muscle-related pathogenesis indeed exists in CVM.

In addition to monogenic etiology, association analyses on rare and novel variants revealed the genetic architecture of CVM. We found that rare variant signals are enriched in embryogenic processes and common variant signals are enriched in metabolism and homeostasis maintenance (also addressed in the responses to the following questions). These findings revealed the different biological mechanisms through which rare and common variants affect the normal development of the spine.

Another important insight into the developmental biology of human spine is achieved through integrated analyses between genetics association signals and single-cell transcriptome of the human embryonic spine. There are 30 different cell types that could participate in the human embryonic spine, and it would be difficult to identify the key cell types without manipulating the embryo, which is inapplicable in humans. We found that rare and common variant signals concordantly ascribed the disease liability to the notochord and myocyte/myoblasts. The myocyte/myoblasts enrichment reiterates our findings from the monogenic analyses. The role of notochord in spinal development has also been a hot topic in recent years⁴⁻⁶. Our findings help further establish its role from the 'model organism' of humans.

We have further articulated these points in the discussion part of the revised manuscript.

Minor comments:

Figure 2e

The exclusive reliance on p-values in GO term enrichment or KEGG analysis is a widespread statistical misconception. It is crucial to present effect sizes in conjunction with statistical significance (lower than alpha level you have settled for this study), as the number of GO terms or KEGG pathways covered by each term/pathway can exhibit significant variations. The p-value, on its own, primarily reflects statistical accuracy, while effect sizes offer a quantitative measure of the practical significance or magnitude of the observed effects for each term or pathway.

Response: Thank you for pointing this out. We agree that we should prioritize pathway enrichment results according to effect sizes rather than p-values. We have updated the prioritization according to the enrichment score, i.e. the ratio of positive hits in each pathway controlled against the size of the pathway.

l.178 - 183

I assume the result could change if the above statistical misconception was solved.

Response: The order of enriched pathways indeed changed after updating the prioritization. However, the major results and conclusion remained unchanged, suggesting the robustness of our analyses. The most enriched Gene Ontology (GO) processes were still associated with the morphogenesis of various organs and apoptosis during development. Calcium signaling, which has the lowest p-value, also presented an enrichment score that ranks prominently high. We have updated the manuscript according to the updated results.

Figure 3c

Same as Figure 2e.

Response: We have updated the enrichment analyses of EWAS signals accordingly.

l.195 - 198

The authors devoted considerable effort to gather the comprehensive dataset presented in this manuscript, and I commend their dedication and diligence. However, this is the main reason why I am concerned with the description and results of current version. Relying only on P-values works fine for EWAS or GWAS analysis, as the sample number used is the same for every locus, however, this does not stand for GO term and KEGG pathway analyses as I explained above.

Response: Thank you for raising this important point. It is indeed important to prioritize and report appropriate statistical results based on the characteristics of the variables. Similar as the enrichment results of burden signals, the order of pathways slightly changed but the major results and conclusion remained unchanged. Pathway enrichment analysis of the 730 EWAS genes pinpointed various metabolic pathways, such as the flavonoid metabolic process and ascorbate and aldarate metabolism. These findings suggest that common variants, with smaller effect sizes than URVs, contribute to CVM through the interference of the homeostasis of the developing spine rather than impairing critical embryogenic processes. We have updated the figures and the descriptions of the pathway enrichment tests for burden and EWAS signals.

Figure 4 and Methods

Please specify how many embryos were used for each developmental stage. While statistical power and accuracy may be compromised if all of these were done only for one sample each (n=1), it is still precious data.

Response: We only have one sample for each developmental stage due to the difficulties of obtaining human embryos. We have updated the information in the Methods section.

References

- 1 Chen, W. *et al.* TBX6 missense variants expand the mutational spectrum in a non-Mendelian inheritance disease. *Hum Mutat* **41**, 182-195 (2020). <https://doi.org:10.1002/humu.23907>
- 2 Chong, J. X. *et al.* Autosomal-Dominant Multiple Pterygium Syndrome Is Caused by Mutations in MYH3. *Am J Hum Genet* **96**, 841-849 (2015). <https://doi.org:10.1016/j.ajhg.2015.04.004>
- 3 Zhao, S. *et al.* Expanding the mutation and phenotype spectrum of MYH3-associated skeletal disorders. *Npj Genom Medicine* **7**, 11 (2022). <https://doi.org:10.1038/s41525-021-00273-x> PMID - 35169139
- 4 Sun, X. *et al.* Dstyk mutation leads to congenital scoliosis-like vertebral malformations in zebrafish via dysregulated mTORC1/TFEB pathway. *Nat Commun* **11**, 479 (2020). <https://doi.org:10.1038/s41467-019-14169-z>
- 5 Gray, R. S. *et al.* Loss of col8a1a function during zebrafish embryogenesis results in congenital vertebral malformations. *Dev Biol* **386**, 72-85 (2014). <https://doi.org:10.1016/j.ydbio.2013.11.028>
- 6 Bagwell, J. *et al.* Notochord vacuoles absorb compressive bone growth during zebrafish spine formation. *Elife* **9** (2020). <https://doi.org:10.7554/eLife.51221>

REVIEWER COMMENTS

Reviewer #3

This is a nicely written, comprehensive and interesting manuscript describing the genetic underpinning of CVM from multiple angles.

The paper is relevant and I have only minor comments (with the #3 being the most important one):

1. It would seem that the variants in MYH3, MYH7, ALPK3, and RYR1 truly identify a phenocopy of CVM, where the primary muscle defect result in secondary vertebral fusion without a developmental spine defect. I would clarify it in the text, it seems a very important distinction for genetic counseling, diagnosis, and management, since it would be conceivable that these vertebral fusions can be prevented or at least mitigated if the diagnosis is made ahead of time.

2. Line 131: what the author mean for "low-term X-ray follow-up"? Is that "long-term"?

3. EWAS: this seems the most preliminary and less informative of the analyses. The results are not striking and even if the top signals exceed a FDR threshold, the common variant framework of the EWAS would still "make it abide" to the GWAS rules where the standard accepted significance threshold is 5×10^{-8} ; moreover, even if restricting to coding variants, the ones with high MAF (as the top signals are) should have a LD pattern that would bring in other follow up signals (with the caveat of the sparse map in exome sequencing but nevertheless). Therefore singletons should be looked at with at least a bit of caution.

Since the authors conducted WGS in the majority of cases, it would be very interesting to extract all common variants from this dataset and conduct a traditional GWAS on common variants using 611 WGS cases and 904 controls. Even with smaller sample size, true signals should be retrieved (hence helping interpreting the EWAS) and by leveraging the full genomes, it might result in true novel associations and also better estimates of heritability.

4. Heritability estimates are likely overestimated being this a discovery study. Please discuss.

5. Please provide PC plots in supplementary information to show genetic matching of the cases and controls

Reviewer #3

This is a nicely written, comprehensive and interesting manuscript describing the genetic underpinning of CVM from multiple angles.

The paper is relevant and I have only minor comments (with the #3 being the most important one):

1. It would seem that the variants in MYH3, MYH7, ALPK3, and RYR1 truly identify a phenocopy of CVM, where the primary muscle defect result in secondary vertebral fusion without a developmental spine defect. I would clarify it in the text, it seems a very important distinction for genetic counseling, diagnosis, and management, since it would be conceivable that these vertebral fusions can be prevented or at least mitigated if the diagnosis is made ahead of time.

Response: Thank you for the suggestion. It is indeed important to distinguish muscle-related vertebral fusion from actual congenital anomalies in clinical practice. We have addressed that point in the Discussion section of the revised manuscript:

“Distinguishing muscle-related vertebral fusion from bona fide malformation of the vertebral column is important for genetic counseling, diagnosis, and disease management. The progressive nature of this distinct clinical entity also prompts the opportunity for early interventions that target myoblasts or myocytes.”

2. Line 131: what the author mean for “low-term X-ray follow-up”? Is that “long-term”?

Response: We apologize for the typo. We have corrected it in the revised manuscript.

3. EWAS: this seems the most preliminary and less informative of the analyses. The results are not striking and even if the top signals exceed a FDR threshold, the common variant framework of the EWAS would still “make it abide” to the GWAS rules where the standard accepted significance threshold is 5×10^{-8} ; moreover, even if restricting to coding variants, the ones with high MAF (as the top signals are) should have a LD pattern that would bring in other follow up signals (with the caveat of the sparse map in exome sequencing but nevertheless). Therefore singletons should be looked at with at least a bit of caution.

Since the authors conducted WGS in the majority of cases, it would be very interesting to extract all common variants from this dataset and conduct a traditional GWAS on common variants using 611 WGS cases and 904 controls. Even with smaller sample size, true signals should be retrieved (hence helping interpreting the EWAS) and by leveraging the full genomes, it might result in true novel associations and also better estimates of heritability.

Response: Thank you for the constructive suggestion.

Firstly, we fully agree that a standard accepted significance threshold of 5×10^{-8} should be set up although no variant reached this threshold in our study. We have addressed this issue in the Results section of the revised manuscript:

“No variant reached a canonical significance threshold of 5×10^{-8} .”

Secondly, we mapped the LD patterns of the three leading signals (revised Supplementary Fig. 9), but did not see follow-up signals with strong LD. We have addressed this point in the Result section of the revised manuscript:

“No significant linkage disequilibrium cluster was detected for these three signals (Supplementary Fig. 9)”

a

b

c

Revised Supplementary Fig. 9

In addition, we utilized the WGS data (535 undiagnosed cases and 892 controls that passed QC) and extracted all variants within $\pm 100\text{kb}$ of the three loci for association analysis. However, the three leading signals are not significant in the WGS analysis (rs3749977, P-value = 0.83; rs139070113, P-value = 0.89; rs1126477, P-value = 0.34), probably due to the lack of power when using this subset of our cohort. We did not find any significant SNPs in LD with these three signals, either. Therefore, we decided not to put the WGS-derived data into the manuscript. Unfortunately, we did not perform a genome-wide analysis for two reasons: a) as the reviewer mentioned, the sample size of our WGS dataset is underpowered for genome-wide discovery, and b) it is out of the scope of this study, which is essentially exome-wide. Finally, we understand the heritability from our EWAS analysis may be overestimated, which is especially true given the fact that the signals are not significant anymore based on a subset of the samples, as also addressed in the next question.

4. Heritability estimates are likely overestimated being this a discovery study. Please discuss.

Response: We fully agree that the heritability is likely overestimated in this study. We have addressed this limitation in the Discussion section in the Discussion section of the revised manuscript:

“Through a preliminary heritability estimation, we identified 3.6% and 9.7% of the heritability for oligogenic and polygenic genetic architecture, respectively. The heritability may be overestimated due to the discovery nature of our study.”

5. Please provide PC plots in supplementary information to show genetic matching of the cases and controls

Response: Thank you for the suggestion. We have included a supplementary figure that how the genetic matching of cases and controls.

Revised Supplementary Fig. 11

References

- 1 Chen, W. *et al.* TBX6 missense variants expand the mutational spectrum in a non-Mendelian inheritance disease. *Hum Mutat* **41**, 182-195 (2020). <https://doi.org:10.1002/humu.23907>
- 2 Chong, J. X. *et al.* Autosomal-Dominant Multiple Pterygium Syndrome Is Caused by Mutations in MYH3. *Am J Hum Genet* **96**, 841-849 (2015). <https://doi.org:10.1016/j.ajhg.2015.04.004>
- 3 Zhao, S. *et al.* Expanding the mutation and phenotype spectrum of MYH3-associated skeletal disorders. *Npj Genom Medicine* **7**, 11 (2022). <https://doi.org:10.1038/s41525-021-00273-x> PMID - 35169139
- 4 Sun, X. *et al.* Dstyk mutation leads to congenital scoliosis-like vertebral malformations in zebrafish via dysregulated mTORC1/TFEB pathway. *Nat Commun* **11**, 479 (2020). <https://doi.org:10.1038/s41467-019-14169-z>
- 5 Gray, R. S. *et al.* Loss of col8a1a function during zebrafish embryogenesis results in congenital vertebral malformations. *Dev Biol* **386**, 72-85 (2014). <https://doi.org:10.1016/j.ydbio.2013.11.028>
- 6 Bagwell, J. *et al.* Notochord vacuoles absorb compressive bone growth during zebrafish spine formation. *Elife* **9** (2020). <https://doi.org:10.7554/eLife.51221>